# Development and evaluation of the ARM algorithm: A novel approach to quantify musculoskeletal disorder risk factors in manual wheelchair users in the real world

Omid Jahanian[1¤]*, Meegan G. Van Straaten[1], Kathylee Pinnock Branford[2], Emma Fortune[1], Stephen M. Cain[2], Melissa M. B. Morrow[3]

1 Division of Health Care Delivery Research, Robert D. and Patricia E. Kern Center for the Science of Health Care Delivery, Mayo Clinic, Rochester, Minnesota, United States of America, 2 Department of Chemical and Biomedical Engineering, West Virginia University, Morgantown, West Virginia, Minnesota, United States of America, 3 Department of Nutrition, Metabolism, & Rehabilitation Sciences, Center for Health Promotion, Performance, and Rehabilitation Research, The University of Texas Medical Branch, Galveston, Texas, United States of America

¤ Current address: Department of Physical Medicine & Rehabilitation, Mayo Clinic, Rochester, Minnesota, United States of America
* jahanian.omid@mayo.edu

## Abstract

This study aimed to develop and evaluate the ARM (arm repetitive movement) algorithm using inertial measurement unit (IMU) data to assess repetitive arm motion in manual wheelchair (MWC) users in real-world settings. The algorithm was tested on community data from four MWC users with spinal cord injury and compared with video-based analysis. Additionally, the algorithm was applied to in-home and free-living environment data from two and sixteen MWC users, respectively, to assess its utility in quantifying differences across activities of daily living and between dominant and non-dominant arms. The ARM algorithm accurately estimated active and resting times (>98%) in the community and confirmed asymmetries between dominant and non-dominant arm usage in in-home and free-living environment data. Analysis of free-living environment data revealed that the total resting bout time was significantly longer (P = 0.049) and total active bout time was significantly shorter (P = 0.011) for the non-dominant arm. Analysis of active bouts longer than 10 seconds showed higher total time (P = 0.015), average duration (P = 0.026), and number of movement cycles per bout (P = 0.020) for the dominant side. These findings support the feasibility of using the IMU-based ARM algorithm to assess repetitive arm motion and monitor shoulder disorder risk factors in MWC users during daily activities.

## Introduction

Manual wheelchair (MWC) users with spinal cord injury (SCI) utilize their upper limbs for both mobility and activities of daily living [1, 2]. As a result, they are extensively exposed to

**Data Availability Statement:** All relevant data are within the manuscript and its Supporting information files.

**Funding:** This publication was made possible by funding from the National Institutes of Health (NIH; grant no. R01HD84423). NIH had no role in study design, data collection and analysis, decision to publish, or preparation of the manuscript.

**Competing interests:** The authors have declared that no competing interests exist.

musculoskeletal disorder risk factors, including increased levels of above-shoulder height arm elevation, repetitive motions, and shoulder loading that accelerate shoulder pathology and pain progression beyond natural aging [1, 3]. Rotator cuff pathology is one of the most common findings when treating shoulder-related disability in this population [1, 4]. The current understanding of rotator cuff tendon pathology emphasizes the significant impact and interplay of repetitive arm motion while performing loaded tasks [5, 6]. However, our knowledge of repetitive motion during free-living daily life is limited which poses a barrier to developing informed preventative programs to protect the shoulders of MWC users.

To better understand and assess the musculoskeletal disorder risk factors that MWC users are exposed to during daily life, real-world data collection is necessary. Inertial measurement units (IMUs), which contain an accelerometer, gyroscope, and magnetometer, can be worn in the free-living environment to continuously capture arm use across a large majority of the day. IMUs have previously been used to quantify wheelchair propulsion behaviors and daily arm use in MWC users [7–12]. In a recent study [13], inspired by ergonomics literature [14], we defined risk and recovery metrics to assess arm use in MWC users in the free-living environment. Various tools have been devised and validated to screen repetitive tasks involving the upper extremities, aiming to assess the risks of work-related musculoskeletal disorders [15–17]. Additionally, methods employing wearable sensors have been developed to objectively measure repetitive upper arm motions [18–20], facilitating the evaluation of physical exposure and musculoskeletal risks in the workplace. In contrast to ergonomic studies that focus on structured and known tasks, our study with MWC users analyzes unsupervised and unstructured free-living daily activities, providing a better reflection of natural habitual behavior. Consequently, the application of ergonomic definitions of repetition and quantitative methods directly to data from daily life is challenging, necessitating alternative metrics that offer a valid interpretation of arm use in a free-living environment.

In this study, we propose an ARM (arm repetitive movement) algorithm based on IMU data and informed by studies of ergonomics to interpret repetitive arm motion in MWC users with SCI [14, 20]. In order to investigate the feasibility of this novel approach to measure the shoulder musculoskeletal disorder risk factors that MWC users are exposed to in the real world, this study aimed to (1) apply the ARM algorithm to supervised community environment activity data from MWC users with SCI to quantify the differences between the criterion-standard video-based and sensor-based analyses of repetitive arm motion, and (2) apply the ARM algorithm to supervised in-home activity data and unsupervised free-living environment activity data from MWC users with SCI to demonstrate the utility of the metrics for quantifying differences across real-world tasks and dominant versus non-dominant limbs.

## Methods

### Participants and data collection

This study was approved by the Mayo Clinic Institutional Review Board. Participants were recruited as part of a larger longitudinal study, "Natural History of Shoulder Pathology in Wheelchair Users" (NCT02600910). Individuals between the ages of 18 and 70 with SCI who use a MWC as their primary mode of mobility were recruited and written informed consent was obtained from them. Inclusion criteria included functional upper extremity range of motion, defined as active shoulder flexion and abduction of at least 150˚ and the ability of the participant to touch the opposite shoulder, the back of his/her neck and his/her low back. Participants were excluded if they self-reported a previous diagnosis of rotator cuff tendon tear. Additionally, MWC users were excluded if they had health complications that would inhibit

their ability to participate in the larger longitudinal study such as stage IV pressure injuries, extensive comorbidities, or severe pain. Detailed inclusion criteria are described elsewhere [12].

## Algorithm development

**Calculation of humeral elevation.** Participants were asked to wear three IMUs (Opal or Emerald, APDM Inc., Portland, Oregon), one on each upper arm and one on their chest. Additionally, participants were asked to perform a functional calibration at the beginning of data collection [8]. Our established method was used to calculate the humeral elevation angle [8, 12]. In summary, a sensor fusion method (Kalman filter) was employed to estimate IMU orientation from linear acceleration and angular velocity [21]. Custom MATLAB code (MathWorks, Natick, MA) was developed to calculate orientations of anatomical axes relative to IMU-fixed reference frames using data collected during each participant's functional calibration postures and movements [8]. The orientation of the upper arm in an inertial (world) reference frame was then estimated using the orientation of the IMU and the orientation of the anatomical axes relative to the IMU-fixed reference frame. The humeral elevation angle was defined as the angle between the long axis of the body segment and the vertical. These angles depend solely on the estimated direction of gravity relative to the body segment, making them drift-free metrics for quantifying body segment motions [8]. The calculated humeral elevation angles range between 0–180˚, with 0˚ indicating the arm was down and perfectly aligned with gravity, and 180˚ indicating the arm was raised overhead and aligned with gravity. These methods have previously been validated in unpublished data involving five individuals with SCI who completed 10 reaching tasks. In comparison to the gold standard (electromagnetic system), the absolute error and percentage of error for the range of motion were found to be -0.06±1.12˚ and -1.44±1.28%, respectively. Similarly, for the maximum elevation achieved during each reach, the absolute error and percentage of error were 2.59±2.47˚ and 2.04±2.47%, respectively.

**Defining repetition and active and resting bouts.** Repetition: To define repetition in the free-living environment, first the basic pattern of one arm movement cycle was identified. One movement cycle was defined as an arm elevation greater than a movement threshold (for example 10˚), followed by an arm lowering greater than the movement threshold. Movements with smaller ranges of motion than the movement threshold that occurred between successive elevation and lowering events were termed 'idle'. Idle periods were treated as a part of the movement cycle (Fig 1a). A range of thresholds were investigated to determine the optimal movement threshold.

Active and Resting Bouts: Next the intervals (time) of no activity between the consecutive movement cycles were calculated. An activity bout contains one or more movement cycles that may be separated by no activity intervals up to 7 seconds. A no activity interval of greater than 7 seconds marks the end of an activity bout [22], (Fig 1b). Seven seconds was chosen in a study specific to wheelchair propulsion by Tolerico et al. as the amount of time with no wheelchair activity in which the participant was considered not-active [22], and that "resting" time frame was also used in this study. The intervals of no activity greater than 7 seconds were considered resting bouts (Fig 1c). Finally, the number of movement cycles per activity bout and the kinematic and temporal features of each cycle and each activity bout were calculated (Table 1). Metrics were calculated for all active bouts as well as active bouts that were longer than 10 seconds.

## Data collection, processing, and statistical analysis

Three data sets were used in this study. Data sets were: (1) a supervised community data collection (n = 4) in which data were collected in the outpatient clinic setting (parking ramp,

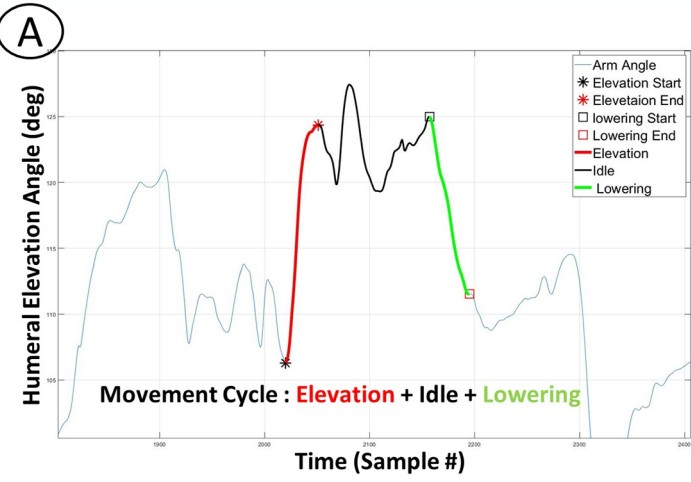

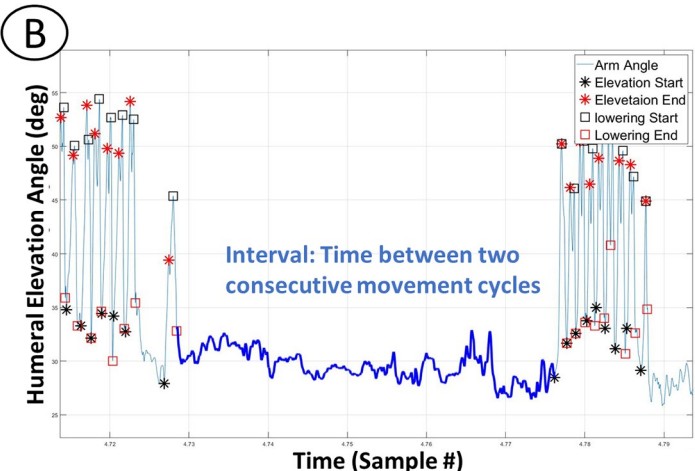

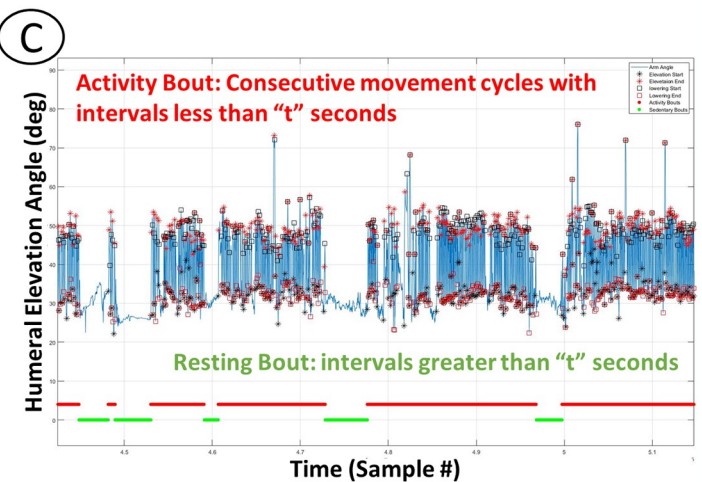

**Fig 1. Definition of active and resting bouts based on humeral elevation data.** (A): Illustrates the definition of a movement cycle, including an Elevation event, a Lowering event, and Idle time. (B): Illustrates an interval between two consecutive movement cycles. (C): Depicts active bouts (wheelchair propulsion) and resting bouts (intervals greater than 7 sec) in red and green horizontal bars, respectively.

**Table 1. Resting and activity parameters of interest and their definitions.**

| Metric | Definition |
|---|---|
| Number of resting bouts | The total number of resting bouts during the data collection time |
| Total resting time | The sum of all periods of resting bouts during the data collection time |
| Resting bout duration | The average length of resting bouts during the data collection time |
| Number of active bouts | The total number of active bouts during the data collection time |
| Total active time | The sum of all periods of active bouts during the data collection time |
| Active bout duration * | The average length of active bouts during the data collection time |
| Number of movement cycles * | The average number of movement cycles per each active bout during the data collection time |
| Humeral elevation angle * | The average median humeral elevation angle for each active bout across all active bouts |
| Peak humeral elevation angle * | The average peak humeral elevation angle for each active bout across all active bouts |

*: these metrics were calculated for all active bouts as well as active bouts longer than 10 seconds.

medical building) with synchronized video data (S2 Appendix; in January and February 2020), (2) a supervised in-home data collection (n = 2; in March and April 2022), and (3) a free-living, unsupervised longer (> 10 hours) data collection (n = 16; from July 2017 to March 2019). These data sets are detailed in Table 2. Analyses of the data were conducted to (1) evaluate the sensitivity of the proposed approach to the threshold choices (for arm elevation and lowering) and quantify the differences between the criterion-standard video-based and sensor-based (proposed approach) analyses of repetitive arm motion in a supervised community environment in MWC users with SCI, and (2) demonstrate the utility of the proposed approach for quantifying differences across real world tasks, using the supervised in-home data, and the differences in arm use between dominant and non-dominant limbs in the free-living environment (unsupervised).

**Community environment data collection, processing, and analysis.** Video and IMU data were collected from 4 MWC users with SCI (Table 2; S2 Appendix). One researcher (OJ, with greater than 9 years of movement analysis experience) rated the video data as resting, propulsion activity, or non-propulsion activity by determining the starting and ending times for each activity (excluding small fidgeting movements). The rated video data (propulsion activity

**Table 2. Description of the data sets.**

| Data Set | Participants | Data Collection |
|---|---|---|
| Supervised community data collection | 4 males, Age: 32 ± 3 years<br>Time Since Injury: 5 ± 2 years<br>Injury Level: T5-L1 | MWC-based activities including level and inclined propulsion, car transfer, cross body lifting, and reaching in a community environment. |
| Supervised in-home data collection | 2 MWC users with SCI<br>Participant A: male, Age: 29 years<br>Time since injury: 3 years<br>Injury Level: T9-T10<br>Participant B: female, Age: 39 years Time Since Injury: 7 years<br>Injury Level: T4 | Daily routine activities requiring awkward postures, loading, and repetition: dressing, transfers, indoor MWC propulsion, vacuuming/sweeping floor and meal preparation. |
| Unsupervised free-living data collection | 16 MWC users with SCI, 14 males Age: 41 ± 3 years<br>Time Since Injury: 11 ± 12 years<br>Injury Level: C6-L1 | Free-living data collection. Participants instructed not to change their typical daily routine. |

and non-propulsion activity) were used as criterion standard data to evaluate the accuracy of the estimated active and resting times based on four different movement thresholds (5˚, 10˚, 15˚, 20˚). We selected these movement thresholds based on ergonomic literature, particularly the study conducted by Thamsuwan et al. [20], which used accelerometers to characterize upper arm repetitive motions. Their study revealed that increasing the threshold (5˚ to 30˚) led to a decrease in calculated repetition rates, with significant differences observed at 15˚ to 30˚.

To evaluate the applicability of this approach for characterization of arm use during active bouts, the number of MWC propulsion stroke cycles, as an exemplar of repetitive arm activities, was estimated using the ARM algorithm with three different movement thresholds (5˚, 10˚, 15˚) and compared with the actual number of stroke cycles from video data (propulsion activity). We deliberately omitted the use of a 20˚ threshold due to our observations, indicating a limited range of motion in the upper arm during MWC propulsion, especially on hard, level surfaces.

**In-home data collection, processing, and analysis.** Participants were interviewed to determine which activities were perceived to be repetitious, use awkward postures and were physically demanding. Lower body dressing, in-home transfers, floor cleaning, and meal preparation were then chosen for data collection. Participants were asked to do portions of these tasks during this data collection. For example, they cleaned the floor for a few minutes rather than an entire room. Video and IMU data were collected from 2 MWC users with SCI (Table 2). The ARM algorithm with the movement threshold that provided the highest performance for estimating the number of movement cycles during repetitive arm activities in the community environment was applied to the in-home IMU data. The calculated metrics were used to identify both participants' active bouts of dominant and non-dominant arm use during supervised daily activities.

**Free-living environment data collection, processing and analysis.** Participants (16 MWC users with SCI, Table 2; S1 Appendix) were provided with the IMUs and requested to don the sensors for two typical days first thing in the morning, wear them for all daily activities except bathing/swimming, and take them off before bed. Additionally, participants were asked to perform a functional calibration at the beginning of each day and after re-donning the sensors if they were removed for any reason during the day [8]. The ARM algorithm was applied to the IMU data, and the calculated metrics were reported and used to explore within-participant differences in arm use between dominant and non-dominant arms. All parameters of interest (Table 1) were calculated and compared between dominant and non-dominant arms. Due to the sample size and the non-normal distribution of the data, the statistical analyses were conducted using separate Wilcoxon Signed-Rank tests in SPSS 28. A P-value less than 0.05 was considered statistically significant.

## Results

### Community environment data

The rated video data indicated that the total time of data collection for all participants (N = 4) was 3123 seconds (52 minutes). Participants were active for a total of 1592 seconds (26 minutes) and total resting time was 1531 seconds (25 minutes). Comparisons of model estimates of active time and sedentary time (dominant side) using four different movement thresholds (5˚, 10˚, 15˚, 20˚) to criterion standard data revealed that employing movement thresholds of 5˚ and 10˚ in the proposed sensor-based approach produced results comparable to those obtained through the criterion standard video-based analysis (Table 3). Model estimates of the number of MWC propulsion stroke cycles when implementing a 10˚ movement threshold had

**Table 3. Accuracy measures of the ARM algorithm in estimating the active time and resting time for the dominant arm.** Accuracy is the ratio of true active and true resting to the total time, sensitivity is the ratio of true active to the total active time, and specificity is the true resting to the total resting time.

| Measures | 5 Deg | 10 Deg | 15 Deg | 20 Deg |
|---|---|---|---|---|
| **Accuracy** | 0.995 | 0.984 | 0.939 | 0.761 |
| **Sensitivity** | 0.995 | 0.969 | 0.880 | 0.527 |
| **Specificity** | 0.996 | 0.999 | 0.999 | 0.999 |

**Table 4. Estimated number of stroke cycles when using different movement thresholds (5–15 degrees).** Negative errors indicate underestimation and positive errors indicate overestimation.

| | Actual | 5 Deg | 10 Deg | 15 Deg |
|---|---|---|---|---|
| **Number of Stroke cycles** | 1116 | 1204 | 1091 | 780 |
| **Error** | N/A | 8% | -2% | -30% |

the lowest error ($< 2\%$ underestimation error; Table 4), and, therefore, 10° was selected as the optimal threshold for the application of the ARM algorithm to the In-Home and Free-Living Environment data. Movement thresholds smaller than 10° led to overestimation, and greater than 10° led to underestimation of the number of push counts.

## In-home data

The total in-home data collection time while performing daily routine activities, including lower body dressing simulation, transfers, indoor MWC propulsion, vacuuming/sweeping and meal preparation, was 2465 seconds (41 minutes) and 1099 seconds (18 minutes) for participant A (female) and participant B (male), respectively. Table 5 indicates longer active time and shorter resting time for the dominant arm compared to the non-dominant arm during in-home activities for both participants. Table 6 indicates the counts of movement cycles, along with the median and peak humeral elevation angles during daily activities. These measurements signify repetition and awkward postures, recognized as key extrinsic risk factors for musculoskeletal disorders in the ergonomics literature.

**Table 5. Active and resting bouts information for dominant (Dom) arm and non-dominant (Non-Dom) arm during in-home activities.**

| | Participant A | | Participant B | |
|---|---|---|---|---|
| | **Dom Arm** | **Non-Dom Arm** | **Dom Arm** | **Non-Dom Arm** |
| **Total Data Collection Time (Sec)** | 2465 | 2465 | 1099 | 1099 |
| **Active Bouts** | | | | |
| Total Time (Sec) | 1292 | 1026 | 756 | 572 |
| Number of Bouts | 76 | 57 | 36 | 26 |
| Duration (Sec) | | | | |
| Mean (SD) | 17 (25) | 18 (26) | 21 (28) | 22 (13) |
| Median (IQR) | 10 (11) | 13 (14) | 13 (12) | 21 (17) |
| **Resting Bouts** | | | | |
| Total Time | 1173 | 1439 | 343 | 527 |
| Duration (Sec) | | | | |
| Mean (SD) | 17 (13) | 20 (16) | 17 (16) | 25 (22) |
| Median (IQR) | 12 (9) | 14 (16) | 11 (12) | 15 (25) |

**Table 6. Upper arm elevation and repetition data for dominant (Dom) arm and non-dominant (Non-Dom) arm during in-home activities.**

| Activity | Participant A | | Participant B | |
|---|---|---|---|---|
| | Dom Arm | Non-Dom Arm | Dom Arm | Non-Dom Arm |
| **Dressing** | | | | |
| **Duration (Sec)** | 219 | 219 | 204 | 204 |
| **# Movement Cycles** | 40 | 34 | 44 | 45 |
| **Median Humeral Elevation Angle per cycle** | | | | |
| Mean (SD) | 46 (16) | 49 (18) | 44 (14) | 43 (15) |
| Median (IQR) | 44 (12) | 48 (15) | 43 (14) | 43 (18) |
| **Peak Humeral Elevation Angle per cycle** | | | | |
| Mean (SD) | 59 (20) | 67 (27) | 62 (17) | 62 (18) |
| Median (IQR) | 54 (20) | 64 (23) | 58 (24) | 61 (23) |
| **Propulsion and Maneuvering** | | | | |
| **Duration (Sec)** | 28 | 28 | 17 | 17 |
| **# Movement Cycles** | 16 | 15 | 11 | 9 |
| **Median Humeral Elevation Angle per cycle** | | | | |
| Mean (SD) | 40 (13) | 36 (10) | 38 (14) | 36 (11) |
| Median (IQR) | 38 (16) | 33 (13) | 37 (25) | 35 (16) |
| **Peak Humeral Elevation Angle per cycle** | | | | |
| Mean (SD) | 54 (13) | 50 (8) | 49 (10) | 52 (8) |
| Median (IQR) | 49 (14) | 47 (13) | 50 (13) | 52 (9) |
| | Vacuuming | | Sweeping | |
| **Duration (Sec)** | 190 | 190 | 180 | 180 |
| **# Movement Cycles** | 61 | 52 | 90 | 28 |
| **Median Humeral Elevation Angle per cycle** | | | | |
| Mean (SD) | 39 (15) | 37 (17) | 45 (14) | 46 (12) |
| Median (IQR) | 38 (16) | 34 (18) | 48 (16) | 48 (13) |
| **Peak Humeral Elevation Angle per cycle** | | | | |
| Mean (SD) | 53 (15) | 54 (18) | 61 (12) | 57 (13) |
| Median (IQR) | 46 (16) | 46 (18) | 63 (12) | 54 (18) |
| **Meal Preparation** | | | | |
| **Duration (Sec)** | 1498 | 1498 | 480 | 480 |
| **# Movement Cycles** | 181 | 128 | 56 | 33 |
| **Median Humeral Elevation Angle per cycle** | | | | |
| Mean (SD) | 48 (23) | 38 (19) | 58 (24) | 57 (13) |
| Median (IQR) | 36 (41) | 33 (19) | 60 (47) | 58 (12) |
| **Peak Humeral Elevation Angle per cycle** | | | | |
| Mean (SD) | 68 (22) | 59 (22) | 69 (19) | 67 (17) |
| Median (IQR) | 67 (35) | 50 (30) | 68 (34) | 66 (20) |

## Free-living environment data

In the free-living dataset, 11 out of 16 participants had 2 days of usable data and 5 participants had 1 day of data (> 10 hours). The average (SD) duration of data collection was 11.2 (3.1) hours. The results from application of the ARM algorithm to the free-living environment data (Table 7 and Fig 2) indicated that there were some asymmetries for arm use between dominant and non-dominant arms. The total time of the resting bouts was significantly longer for non-dominant arm than dominant arm (P = 0.049), and the total time of active bouts was significantly longer for the dominant arm compared to the non-dominant arm (P = 0.011).

**Table 7. Number of bouts and total time (hours and minutes, hh:mm) for resting and active bouts of dominant (Dom) and non-dominant (Non-Dom) arms during free-living environment data collection from 16 MWC users with SCI.**

| | Number of Bouts | | | Total Time (hh:mm) | | |
|---|---|---|---|---|---|---|
| | Dom Arm | Non-Dom Arm | P | Dom Arm | Non-Dom Arm | P |
| **Resting Bouts** | | | | | | |
| **Mean (SD)** | 339 (184) | 329 (167) | 0.093 | 8:29 (2:32) | 8:42 (2:37) | 0.049 |
| **Median (Q1, Q3)** | 322 (244, 477) | 321 (233, 430) | | 8:29 (7:26, 9:39) | 8:53 (7:52, 10:02) | |
| **All Active Bouts** | | | | | | |
| **Mean (SD)** | 340 (184) | 330 (167) | 0.093 | 2:14 (1:19) | 2:00 (1:08) | 0.011 |
| **Median (Q1, Q3)** | 323 (245, 478) | 322 (234, 431) | | 2:02 (1:41, 2:27) | 1:51 (1:28, 2:20) | |
| **Active Bouts > 10 seconds** | | | | | | |
| **Mean (SD)** | 218 (118) | 208 (110) | 0.114 | 2:01 (1:13) | 1:47 (1:03) | 0.015 |
| **Median (Q1, Q3)** | 198 (163, 285) | 195 (152, 254) | | 1:52 (1:32, 2:17) | 1:38 (1:20, 2:02) | |

Analyzing the activity parameters during active bouts that were longer than 10 seconds indicated that the total time (P = 0.015), average duration (P = 0.026), and average number of cycles per active bout (P = 0.020) were significantly higher for dominant side than non-dominant side.

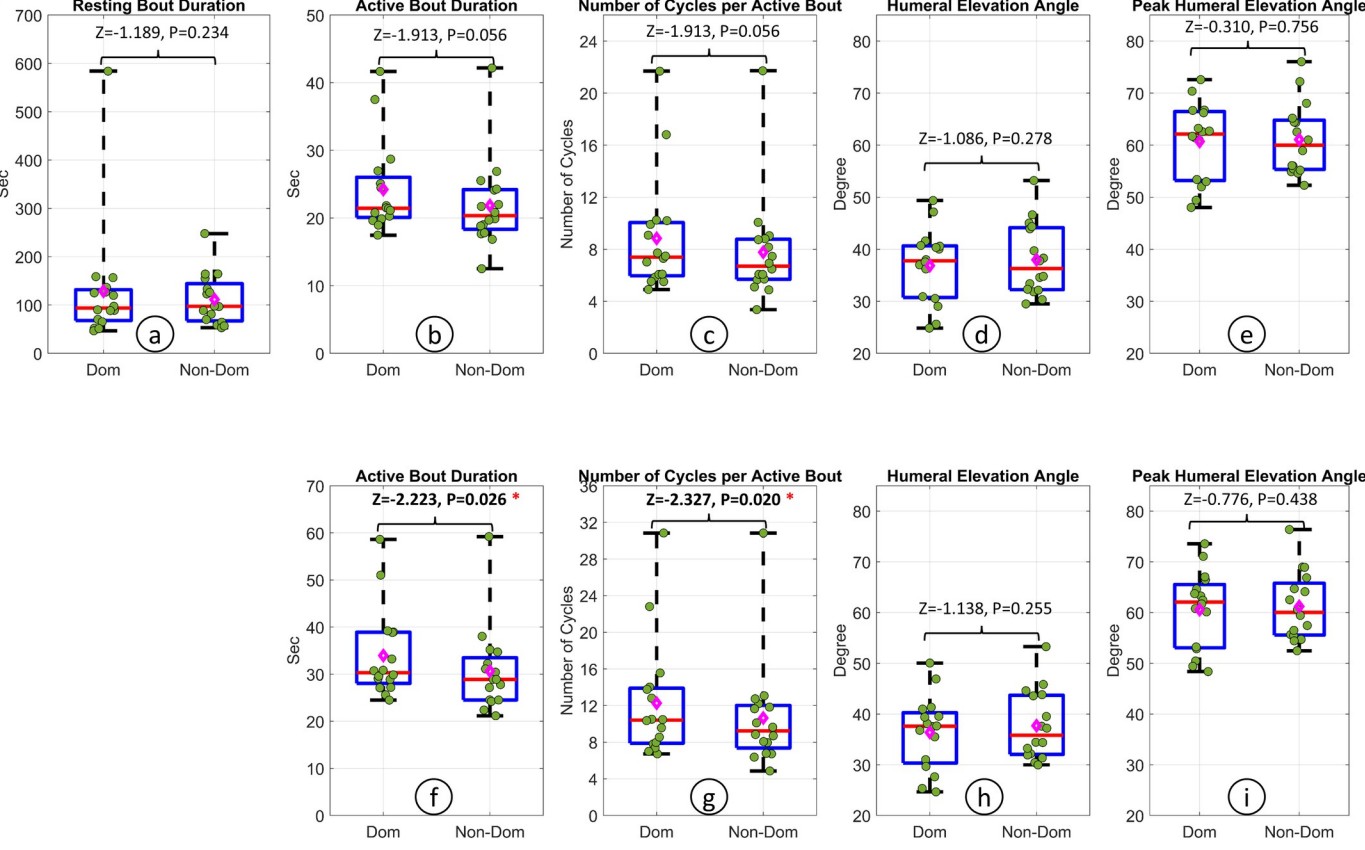

**Fig 2. Box plots for average duration of resting bouts.** (a), average duration, average number of movement cycles per active bout, average median humeral elevation angle, and average peak humeral elevation angle for all active bouts (b-e) and for only active bouts that were longer than 10 seconds (f-i), and statistical results for comparing the parameters between dominant (Dom) and non-dominant (Non-Dom) arms. Data collection from 16 MWC users with SCI during free-living.

## Discussion

In this study, we proposed a novel approach (ARM algorithm) based on upper-arm IMU data, informed by studies of ergonomics, for quantitative assessment of extrinsic factors of shoulder use in MWC users with SCI. We aimed to quantify active bouts within the context of repetitive arm motion and relative bouts of rest in this population. Accuracy and sensitivity analyses of the supervised community-based data indicated that using movement thresholds of 5˚ and 10˚ in the proposed sensor-based approach yielded similar results for estimating active and resting time to those obtained from the criterion-standard video-based analysis. The proposed approach had its highest performance for estimating the number of arm movement cycles during the repetitive arm activity of MWC propulsion when implementing a 10˚ movement threshold.

### In-home data collection: Active and resting time

The results from the in-home data collection demonstrated the feasibility of the proposed approach for quantitative assessment of exposure to repetition and arm elevation in real world settings, in which both are among the risk factors for rotator cuff pathology [6]. Application of the ARM algorithm (with a 10˚ movement threshold) to the in-home data, for two MWC users, demonstrated for both participants that, not surprisingly, their dominant arm was active for a greater time than their non-dominant arm. The average duration of active bouts was similar for both sides. The total resting time was notably longer for the non-dominant arm than the dominant arm.

### In-home data collection: Movement cycles and humeral elevation for dressing and propulsion

The results for the in-home activities indicated that both participants used their arms nearly symmetrically for performing tasks such as dressing and propulsion. The differences in the number of movement cycles between the dominant and non-dominant arms during MWC propulsion can be attributed to the fact that participants utilized their dominant arm for other tasks, such as door handling and turning on/off the light while performing the propulsion task. Lower body dressing exhibited the largest humeral elevation angle compared to the other tasks. Considering that lower legs are lifted, often with longer lever arms away from the body, it implies that the shoulders are being loaded at these elevation angles. This may have important implications for understanding the mechanisms of shoulder pathology.

### In-home data collection: Movement cycles and humeral elevation for vacuuming and meal preparation

Upper arm postures were nearly the same for the dominant and non-dominant arm in both participants during vacuuming and sweeping tasks. However, the number of movement cycles were notably higher for the dominant side which indicates, as expected, that the dominant arm is used more in floor cleaning. Both participants mainly used their dominant arm for preparing meal and the number of cycles were notably higher for the dominant side in comparison to non-dominant side. The median and peak humeral elevation angle per cycle were notably higher for the dominant side in only one of the participants (participant A) during meal preparation. This might be due to the differences in the difficulty level of the recipe. The meal that participant A prepared was more demanding and required higher cooking skills, such as vegetable chopping, while participant B picked a recipe requiring fewer steps and ingredients. Additionally, participant B sat higher in her wheelchair compared to the kitchen

counter height and had better trunk control, which may have led to greater ease in reaching and handling kitchen utensils and ingredients during the cooking process. It is important to mention that we cannot draw any conclusions regarding differences in the number of movement cycles between activities since we only collected a portion of these activities. This metric is included in Table 6 as a demonstration of the type of information that can be calculated with the ARM algorithm.

### Free-living environment data collection

Application of the ARM algorithm to the free-living environment data demonstrated the potential of this approach for interpretation of active bouts, resting bouts, and repetitive motion in unsupervised data collections in MWC users. Consistent with the supervised in-home data, the dominant arm was active for longer durations than the non-dominant arm. The results for the active bouts that were longer than 10 seconds revealed a significantly higher repetition of humeral elevation (number of movement cycles) on the dominant side in comparison to the non-dominant side (the average number of movement cycles per active bout was 12 for the dominant side and 10 for the non-dominant side). These findings demonstrate the expected asymmetry in daily arm usage between the dominant and non-dominant arms among MWC users. This observation provides support for the construct validity of the proposed approach. These findings align with previous studies involving populations with neuro-motor disorders such as people with Multiple Sclerosis, wherein significant asymmetry in the use of upper limbs in real-world situations has been reported [23, 24]. These findings suggest that focusing on active bouts exceeding 10 seconds in duration may yield more meaningful insights compared to analyzing all active bouts indiscriminately. This underscores the need for future studies to establish a clinically relevant threshold for the inclusion of active bouts in the analysis.

The ARM algorithm, proposed in this study, was informed by ergonomics studies, particularly the work conducted by Thamsuwan and colleagues [20]. Their successful development of a method, utilizing accelerometers, to assess repetitive arm motions in apple-harvesting workers served as a foundational reference. The fundamental definition of movement cycles within the ARM algorithm aligns with the repetition cycles identified in Thamsuwan's study, and both studies found comparable thresholds to have the highest accuracy in calculating repetition rates. The utilization of IMUs in the current study provided a better assessment of arm use in MWC users in the free-living environment compared to using only accelerometers, as humeral elevation can be calculated more accurately using both linear acceleration and angular velocity comparted to only linear accelerometer readings. The incorporation of IMUs in the current study offered a more comprehensive assessment of arm usage among MWC users in a free-living environment compared to relying solely on accelerometers. Additionally, the ARM algorithm was employed to evaluate bouts of arm use within the context of repetitive arm movements, providing a holistic analysis of the physical risk factors associated with shoulder musculoskeletal disorders in MWC users.

The small sample size in this study restricted the extent to which these findings can be generalized. Additionally, we can only trust estimates of body segment orientation relative to vertical. Because data collections took place in free-living environments with non-uniform magnetic fields, we did not use magnetometer measurements in our sensor fusion approach for calculating orientations. Therefore, orientation calculations relative to magnetic north are not reliable (i.e., the yaw or heading angle is contaminated by uncorrected integration drift) and are not suitable for accurate calculation of humerothoracic elevation angles and elevation planes relative to the thorax [8, 25]. Therefore, shoulder flexion or abduction were both

interpreted as humeral elevation and are indistinguishable. Due to participant availability or the burden, 1 or 2 days of data were collected for participants, which might not be a reliable representation of their long-term arm use [12, 26].

The results from this study demonstrated that the ARM algorithm is capable of quantifying active bouts, resting bouts, and repetitive motion from unsupervised data collections which could improve assessment of extrinsic factors of shoulder use. Additionally, the proposed approach has the potential to be utilized for the classification of wheelchair propulsion and, consequently, a rough estimation of external forces and loading cycles in MWC users. This study was an essential step towards our long-term research goal to develop a proactive approach for early mitigation of shoulder pathologies in MWC users with SCI and help this population decelerate shoulder soft tissue "aging" while maintaining a full and active lifestyle. However, conducting a validation study with a larger sample size and utilizing optimization algorithms to determine the optimal movement threshold is crucial for ensuring an accurate analysis of musculoskeletal disorder risk factors in MWC users with SCI in the context of repetitive arm motion. Furthermore, the study's limitations, such as having only one expert for video data rating and the absence of intra-rater and inter-rater reliability analyses, should be acknowledged and addressed in future studies. Future studies will include development of risk scores and investigation of the association of these risk scores with the incidence and progression of rotator cuff tendon pathology in MWC users with SCI with a larger sample size and more days of IMU data collection.

## Conclusions

Accurate measurement of free-living IMU data collected from upper arms could provide a more comprehensive understanding of the extrinsic risk factors for accelerated rotator cuff degeneration in MWC users with SCI. The results from this study demonstrate the feasibility and potential of the proposed approach of using the ARM algorithm, based on IMU data, to interpret arm use in the free-living environment and capture repetitive movements and resting periods. This analysis helps in understanding the risk to the shoulder, which can be linked with the development of shoulder pathology. Data collection with larger sample sizes and further task-specific sensitivity analyses are warranted to define motion and rest thresholds that may vary depending on the research question or clinical application.

## Supporting information

**S1 Data. Minimal dataset underlying the results for the free-living environment data collection for the dominant and non-dominant sides.**
(XLSX)

**S1 Appendix. Participant characteristics in free-living environment data collection (N = 16).**
(DOCX)

**S2 Appendix. Experimental set-up.** Examples of experimental set-ups for community data collection (a & b): a) wheelchair propulsion in the community and b) car-to-wheelchair transfer with IMU sensors on upper arms.
(DOCX)

## Author Contributions

**Conceptualization:** Omid Jahanian, Meegan G. Van Straaten, Melissa M. B. Morrow.

**Formal analysis:** Omid Jahanian, Stephen M. Cain.

**Methodology:** Omid Jahanian, Meegan G. Van Straaten, Kathylee Pinnock Branford, Emma Fortune, Stephen M. Cain, Melissa M. B. Morrow.

**Project administration:** Omid Jahanian, Meegan G. Van Straaten.

**Software:** Omid Jahanian, Stephen M. Cain, Melissa M. B. Morrow.

**Supervision:** Melissa M. B. Morrow.

**Visualization:** Omid Jahanian.

**Writing – original draft:** Omid Jahanian, Meegan G. Van Straaten, Melissa M. B. Morrow.

**Writing – review & editing:** Omid Jahanian, Meegan G. Van Straaten, Kathylee Pinnock Branford, Emma Fortune, Stephen M. Cain, Melissa M. B. Morrow.

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
