## [Decision Letter · Decision Letter 0]

3 Nov 2023

PONE-D-23-29198Development and Evaluation of the ARM Algorithm: A Novel Approach to Quantify Musculoskeletal Disorder Risk Factors in Manual Wheelchair Users in the Real WorldPLOS ONE

Dear Dr. jahanian,

Thank you for submitting your manuscript to PLOS ONE. After careful consideration, we feel that it has merit but does not fully meet PLOS ONE’s publication criteria as it currently stands. Therefore, we invite you to submit a revised version of the manuscript that addresses the points raised during the review process.

We look forward to receiving your revised manuscript.

Kind regards,

Jyotindra Narayan

Academic Editor

PLOS ONE

Journal Requirements:

 "This publication was made possible by funding from the National Institutes of Health (grant no. R01HD84423)."

**Additional Editor Comments:**

All the reviewers have provided positive feedback on a study that monitors arm use in manual wheelchair users to assess musculoskeletal risks. As per their suggestions and my carefully reading, I recommend visual enhancements, referencing earlier work, and improved image quality. There are some concerns about result validity, suggesting the inclusion of more experts, clarifications about threshold choices, and additional graphics. Additionally, more clarity on accuracy in arm elevation measurement and socio-demographic information about participants should be highlighted. The authors are suggsted providing demographic information about the study sample and converting Figure 2 into a table.  The authors are finally suggested to address all the concerns raised by the reviwers, especially third reviewer to enhance the study's quality, validity, and presentation. 

Reviewers' comments:

Reviewer's Responses to Questions

**Comments to the Author**

1. Is the manuscript technically sound, and do the data support the conclusions?

Reviewer #1: Yes

Reviewer #2: Partly

Reviewer #3: Yes

2. Has the statistical analysis been performed appropriately and rigorously? 

Reviewer #1: Yes

Reviewer #2: Yes

Reviewer #3: Yes

3. Have the authors made all data underlying the findings in their manuscript fully available?

Reviewer #1: Yes

Reviewer #2: Yes

Reviewer #3: Yes

4. Is the manuscript presented in an intelligible fashion and written in standard English?

Reviewer #1: Yes

Reviewer #2: Yes

Reviewer #3: Yes

5. Review Comments to the Author

Reviewer #1: This study addresses a critical concern, the musculoskeletal risks faced by manual wheelchair users with spinal cord injuries. It offers a commendable approach, quantifying these risks in real-world settings using IMUs to monitor arm use during daily activities. The ARM algorithm, guided by ergonomics, promises crucial insights into repetitive arm motion. This research is a valuable contribution, with potential implications for the development of preventive programs to safeguard the shoulder health of manual wheelchair users.

However, while the research presents an innovative approach and a valuable contribution to the field, there are several critical points that need attention:

1. A pictorial representation or a real environment image of the experimental setup in the methodology section would significantly enhance the clarity and understanding of the study. Relying solely on textual descriptions can make the manuscript less accessible to readers. I strongly recommend the inclusion of such visuals in the sub-section “Participants and Data Collection” line 91.

2. It would be beneficial to incorporate references to relevant earlier work by other researchers in the introduction section before line 80. Highlight the gaps in their research that make this study more impactful. This context can strengthen the paper's academic contribution.

3. The quality of the figures provided in the manuscript is suboptimal and appears to be blurry (Figures 1 and 3) I strongly suggest improving the image quality to ensure that they effectively convey the intended information.

4. A thorough proofreading and editing process is needed to address numerous grammatical, punctuation, and sentence structure errors in the manuscript. Enhancing the overall clarity and readability of the paper is essential.

5. Although the inclusion and exclsuion criteria are described in [12], the authors are strongly suggested to briefly mention the same in the present work to improve the readability of the manuscript. Moreover, the level of SCI for MWC should be clearly highlighted.

6. How four different movement thresholds are selected? Then in lines 153-555, why number MWC propulsion cycles was evaluated with only threee different movement thresholds? The authors have just dumped several pieces of information without adequate explanations. They are suggested to recheck the complete manuscript for such tangential information.

6. The authors are suggested to highlight the critical insights and plausible implications of the observations for In-home and Free Living Environment Data. At present authors have just mentioned the Figures and Tables without any explanations. For example: in Lines 209-211, 'Table 5 demonstrates how the ARM algorithm....task', it is still unclear how?. Similarly, in Lines 220-224, if during long active bouts, the total time and other factors were significantly higher for the dominant side, what's the significance in the context of the MWC users?

8. Finally, the authors are suggested to compare the results of the present work (ARM Algorithm) with state-of-the-art methods, either qualitatively or quantitatively in the Discussion section. Moreover, the limitations of the present work should be explicitly highlighted at the end of the Discussion section.

Reviewer #2: 1. Is the manuscript technically sound, and do the data support the conclusions?

The main issue with the validity of the results is that the ground truth is provided by and dependent on only one expert. Would the addition of another expert result on a different threshold value choice? If that so, would then the (already minimal) differences/size effects become insignificant? The lack of an automated way to calculate the ground truth timings would require providing more experts assessments for consensus.

Other minor issues are:

- What was the rationale for choosing 10 seconds as a long active bout when the threshold was 7 seconds?

- While the effect of drift was mentioned in the Discussion section the manuscript lacks a deeper explanation of the algorithm for humeral elevation angle calculation. Hence, there is uncertainty about the accuracy of the IMU recordings for the long periods in the free-living dataset.

- More graphics (e.g. boxplots, violin plots) could improve understanding of the data and their distribution. Tables could be used as supplementary information.

- There seem to be outliers driving the statistical significance in Fig. 3f and Fig. 3g.

2. Has the statistical analysis been performed appropriately and rigorously?

The statistical analyses are appropriate, although removing outliers should be considered.

3. Have the authors made all data underlying the findings in their manuscript fully available?

All data was made available according to the author’s decalaration.

4. Is the manuscript presented in an intelligible fashion and written in standard English?

More concise writing would improve the manuscript readability. For instance, the text in lines 113-117 was unclear as the resting and active bouts share the same threshold value (one under and another above threshold), but they were referred to as different thresholds, which could be confusing/repetitive.

Line 207 should say “respectivelly” instead of “respectfully”.

Reviewer #3: Relevant topic, nicely written manuscript.

However, some topics need to be addressed/elaborated.

Intro: Clearly written, no comments

Methods:

- it would be good to mention the number of participants included in each portion of the study in the methods section.

- when using a threshold for arm elevation, the accuracy of determining arm elevation angles should be addressed.

Lines 152-156

- wc-propulsion was used as an exemplar of repetitive movements. Were any other movement activities (e.g. above-shoulder height arm elevation) also included? As those movements were mentioned to be risk factors for musculoskeletal disorders in the introduction!

- Could the observer rate the amount of movement of above shoulder movements (e.g. Range of motion) with sufficient accuracy from the 2 dimensional video, to compare with output of the ARM algorithm?? Especially those movements around the thresholds used (5°, 10° and 15°)?

- In line 152 four thresholds are mentioned, in line 155 only three. Which thresholds were actually used for this comparison?

Lines 158 – 167:

- It is completely valid to ask the participants to do only portions of tasks for this specific reason of comparison. Question is how much data should be collected to have a fair comparison. Do you have any information/arguments for such?

- Furthermore, data from two participants seems rather marginal for such comparison and validation. (and yes, I do know video annotation is time consuming, but that shouldn't count as argument not to include more participants, it is about the validity of your method) Please elaborate here.

Results:

The results section could benefit a short description of the study sample, in terms of socio demographics and lesion severity, especially for the unsupervised Community Environment data.

Discussion:

The discussion section mostly discusses results obtained with the ARM method. Although relevant, some more attention should be paid to the strengths and weakness of the ARM method in getting to these results: how valid is the method? When that question is answered, then and only then the results of the application thereof are of interest. Strength and weakness should be addressed in a bit more detail, e.g. with respect to validity on above shoulder arm elevation. eventual a limitation section should be added to the manuscript.

Figures are clear, especially figure 1, explaining the details in the definition of a movement cycle.

Figure 2 should be converted to a compact table, in my modest opinion :-)

6. PLOS authors have the option to publish the peer review history of their article (what does this mean?). If published, this will include your full peer review and any attached files.

Reviewer #1: No

Reviewer #2: No

Reviewer #3: No

---

## [Author Response · Author response to Decision Letter 0]

2 Feb 2024

We appreciate the academic editor and reviewers for their thorough assessment of our manuscript and their valuable feedback. All the provided comments have been carefully addressed, and the manuscript has been revised in accordance with the suggestions made by the reviewers. We are confident that these revisions have significantly enhanced the clarity and overall quality of the manuscript.

---

## [Decision Letter · Decision Letter 1]

27 Feb 2024

Development and evaluation of the ARM algorithm: a novel approach to quantify musculoskeletal disorder risk factors in manual wheelchair users in the real world

PONE-D-23-29198R1

Dear Dr. jahanian,

We’re pleased to inform you that your manuscript has been judged scientifically suitable for publication and will be formally accepted for publication once it meets all outstanding technical requirements.

Kind regards,

Jyotindra Narayan

Academic Editor

PLOS ONE

Additional Editor Comments (optional):

All reveiwers have found the revsied mansucript in a better shape and recommended the same for publication. Congratulations to the authors. 

Reviewers' comments:

Reviewer's Responses to Questions

**Comments to the Author**

1. If the authors have adequately addressed your comments raised in a previous round of review and you feel that this manuscript is now acceptable for publication, you may indicate that here to bypass the “Comments to the Author” section, enter your conflict of interest statement in the “Confidential to Editor” section, and submit your "Accept" recommendation.

Reviewer #1: All comments have been addressed

Reviewer #2: All comments have been addressed

Reviewer #3: All comments have been addressed

2. Is the manuscript technically sound, and do the data support the conclusions?

Reviewer #1: Yes

Reviewer #2: Yes

Reviewer #3: Yes

3. Has the statistical analysis been performed appropriately and rigorously? 

Reviewer #1: Yes

Reviewer #2: Yes

Reviewer #3: Yes

4. Have the authors made all data underlying the findings in their manuscript fully available?

Reviewer #1: Yes

Reviewer #2: Yes

Reviewer #3: Yes

5. Is the manuscript presented in an intelligible fashion and written in standard English?

Reviewer #1: Yes

Reviewer #2: Yes

Reviewer #3: Yes

6. Review Comments to the Author

Reviewer #1: This study represents a significant advancement in the field of rehabilitation and assistive technologies, offering a novel, real-world applicable solution to assess and monitor repetitive arm motion in manual wheelchair users. The ARM algorithm's development and its successful application in quantifying activity differences and arm usage asymmetries highlight the potential to improve intervention strategies for preventing shoulder disorders. The rigorous comparison with video-based analysis and the algorithm's high accuracy in various environments underscore its value as a practical tool for clinical and daily monitoring. Upon thorough evaluation, the revised manuscript has been found to significantly contribute to its respective field. The efforts to address the previously raised concerns have resulted in a substantial improvement in the quality of the work. The clarity and depth presented are commendable, reflecting a dedicated effort to enhance the overall quality of the manuscript. The manuscript is recommended for acceptance.

Reviewer #2: The authors addressed all the comments. They also improved the manuscript considerably in most of its sections. While some comments were addressed by the adding of information, figures, or a rewording of the text, they also specified when a limitation was to be acknowledged. Further clarifications of their methods were also added, which is commendable.

Reviewer #3: I would like to thank the aurthors for this thorough revision, answering all my and other reviewers questions satisfactory !!

7. PLOS authors have the option to publish the peer review history of their article (what does this mean?). If published, this will include your full peer review and any attached files.

Reviewer #1: **Yes: **SUBHASH PRATAP

Reviewer #2: No

Reviewer #3: No

---

## [Editor Report · Acceptance letter]

22 Mar 2024

PONE-D-23-29198R1 

PLOS ONE

Dear Dr. Jahanian, 

I'm pleased to inform you that your manuscript has been deemed suitable for publication in PLOS ONE. Congratulations! Your manuscript is now being handed over to our production team.

Kind regards, 

on behalf of

Dr. Jyotindra Narayan 

Academic Editor

PLOS ONE